## Research Article

adolescents; education; high school; internet; mental health; online information seeking

**Corresponding author:**
Tatjana Gazibara;
Emails: tatjanagazibara@yahoo.com;
tatjana.gazibara@med.bg.ac.rs

# Factors associated with online information seeking about mental health among high school students in Belgrade, Serbia

Tatjana Gazibara ⬡, Jelena Cakic, Milica Cakic, Anita Grgurevic and Tatjana Pekmezovic

Institute of Epidemiology, Faculty of Medicine, 11000 Belgrade, Serbia

## Abstract

Adolescents may not necessarily have a specific mental health challenge to seek information on mental health. They may be genuinely curious on how to better understand these issues, especially when mental health is being discussed in school, among peers and with parents. The purpose of this study was to examine the frequency and factors associated with online information seeking about mental health among adolescents. A total of 702 high school students from Belgrade, Serbia, participated in the study and filled in an anonymous questionnaire about sociodemographics, digital behaviors and the Electronic Health Literacy Scale (eHEALS). The prevalence of seeking information about mental health in our study sample was 23.5% (165/702). The multivariate model showed that having a lower school performance, lower eHEALS score and browsing health blogs, social media and websites run by physicians and health institutions were independently associated with online information seeking about mental health. Additionally, searching for online information about psychoactive substances, bullying and medications was independently associated with online information seeking about mental health among adolescents. Adolescents are familiar with a variety of sources of online health information, but choose specific online platforms to read about mental health. These platforms could be utilized to promote mental well-being in high schools.

## Impact statement

In efforts made by the government and institutions to raise awareness and enhance mental health literacy, adolescents might be interested in expanding their understanding of mental health by using the Internet, regardless of whether or not they encounter mental health challenges. The present study found that one in four high school students seeks online information about mental health. These adolescents were more likely to be females, have lower school performance and have lower e-health literacy levels. They are more likely to use health blogs, social media and websites run by physicians and health institutions, as well as seek information about psychoactive substances, bullying and medications. Websites and Internet platforms can use these study findings to adjust and adapt information delivery regarding mental health to optimize comprehension and knowledge retention.

## Introduction

Mental health challenges, such as emotional, behavioral or eating disorders, affect 13% of teenagers (World Health Organization [WHO], 2021; Watling et al., 2021) and, therefore, represent the key health concern in this population group worldwide. Mental health difficulties in adolescence often remain unrecognized due to the widespread social stigmatization, negative attitudes about help seeking, lack of knowledge about mental health and lack of access to health care providers (WHO, 2021; Leijdesdorff et al., 2021). To increase global mental health literacy, disseminating information is essential in efforts to pave the way for a paradigm shift. This was confirmed by a recent systematic review, which suggests that media campaigns about mental health contribute to positive changes in awareness and stigma reduction, and promote help-seeking (Tam et al., 2024).

In the Republic of Serbia, over the past years, much effort has been made to raise awareness about mental health and prevention of mental health challenges, primarily through the National program for mental health protection endorsed by the Ministry of Health (Ministry of Health, 2019) and UNICEF (UNICEF Serbia, 2021). The peer education network on mental health has been active in primary and secondary schools, while guidebooks for teachers and parents have been developed. Several online platforms (e.g., https://svejeok.rs/sos-centar/) offering

information on how to find help in crisis have been opened. These free resources are accessible via telephone, and one center for children and adolescents (https://www.116111.rs/) also offers access through chat in the evenings.

In circumstances where young people are supported to address their challenges, but also learn more about them, it is not surprising that adolescents search for additional information on the Internet simply because it is available at all times, information is abundant and it is cost-free. Pohl et al. (2024) suggested that adolescents who search for online information about mental health may be classified into those who are affected by emotions, such as stress, anxiety or anger, and those who are curious/wanting to learn. Thus, adolescents may not necessarily have a specific mental health challenge to seek online information about mental health. They may be genuinely curious to better understand these issues, especially when mental health is being discussed in school, among peers and with parents (Pohl et al. 2024). As "digital natives" (i.e., people who have used the Internet from a very early age), adolescents browse various websites, including health forums, social media and blogs, in search of information about mental health (Pretorius et al., 2019) and may have preferences for what sources of information best suit them and bring them closer to the integration of new knowledge.

Bearing in mind that mental health promotion has been one of the strategic goals of the Serbian government to provide support to those in need and raise mental health literacy, it is expected that there will be an increasing number of adolescents who would want to obtain more information about mental health. At the same time, mental health professionals may not always be available to provide information to those adolescents who do not need actual health care services (Gregoric Kumperscak et al., 2020), but desire to obtain additional information. In those circumstances, the Internet can be a helpful resource for the public at large to deepen their knowledge of mental health matters.

With various websites on the Internet, it is sensible to identify specific platforms that contribute to adolescents' seeking of information about mental health on the Internet to recognize the principal sources of information for young people and their preferences. Additionally, it would be informative to have an insight into the patterns of interest in health topics that adolescents seek alongside mental health. The purpose of this study was to examine the frequency and factors associated with online information seeking about mental health among adolescents.

## Methods

This cross-sectional study is part of a larger research study on digital consumer behaviors and online information seeking among adolescents (Gazibara et al., 2020). It was carried out in December 2016 and January 2017. To better understand what health topics adolescents are interested in, the research team has identified a list of various web pages used by consumers online (Sebelefsky et al., 2015) and a list of the most commonly sought health topics by adolescents, as evidenced by previous studies (Kanuga and Rosenfeld, 2004; Martinović et al., 2023). Participants' demographic data were collected as well. For the purpose of this study, responses of students regarding the interest in searching for information on mental health were analyzed through the lens of their demographic data, use of different web pages and other health-related topics that were sought in parallel to mental health information. In this way, it was possible to obtain a comprehensive picture of the characteristics of adolescents who seek information on mental health on the Internet.

### Setting

Secondary education in Serbia is organized and delivered through three programs: high schools, vocational schools and secondary art schools. The vast majority of secondary education institutions are funded by the government, that is, no tuition fees are required. Typical age at high school enrolment is 15 years, and at high school graduation is 19 years. To graduate from high school, students must pass 4 study years as well as the graduation exam at the end of the fourth year. Almost all high schools have two distinctive study tracks: science–mathematics and humanities–languages, which are chosen at enrollment. After high school graduation, students usually continue with their education at universities, whereas students who graduate from vocational schools and secondary art schools usually do not.

### Ethical approval

Ethical approval to conduct the study was provided by the Institutional Review Board of the Faculty of Medicine, University of Belgrade. As the majority of students in the study sample were aged <18 years, their parents were informed about the study through the school board. Opting out of the study was possible; however, none of the parents refused participation. The assent from all minors was provided by accepting to fill in the questionnaire. The consent from all students aged 18 and 19 years was provided by accepting to fill in the questionnaire.

### Recruitment of study participants

A total of 702 high school students from Belgrade, Serbia, participated in the study. Students were recruited from four high schools located in Belgrade. In the Belgrade metropolitan area, there are 21 high schools. In this way, we included approximately one-fifth of all high schools in the city. The procedure of high school selection involved the preparation of paper slips each with the name of the school written on a separate paper slip. Paper slips with school names were folded two times and placed in a nontransparent container. A member of the research team who did not participate in the preparation of paper slips was assigned to pick paper slips from the container to ensure the random selection of high schools. In each high school, classrooms were selected so that students in all 4 study years were included.

To estimate the appropriate sample size, we performed a sample size calculation via the Raosoft calculator (Raosoft, 2016). The following parameters were taken into consideration: margin of error of 5%, confidence interval of 95%, adolescent population size in Belgrade of ~20,000 and population distribution of 50%. A total of 655 students were estimated as an adequate study sample. This sample size was increased by 10% to account for potential students who refuse to participate.

The research team contacted school representatives and had an appointment to present the purpose of the study. The questionnaires were sent to school boards for approval. The approved questionnaires were distributed to students during classes. One member from the research team presented the questionnaire at the beginning of the class, while the teacher was present in the classroom. Members of the research team were at disposal for any clarifications and comments. A similar proportion of participants was selected from each high school: 172 (24.5% from the first high school), 178 (25.3% from the second high school), 169 (24.1% from the third high school) and 183 (26.1% from the fourth high school).

After having introduced the study in front of the class, all high school students who were invited to participate also agreed to take

part in the study (response rate 100%). Such a high response rate could be open to social acceptability bias, that is, acceptance to participate in a survey among hesitant individuals or those who might want to decline participation, because their peers (classmates) are willing to do (Harrison et al., 2015). Students filled the questionnaires inside their classrooms, and every student had a desk to read and respond to the questionnaire items independently. Students were free to quit filling in the questionnaire at any moment and were not coerced to participate.

### Collection of data

This study used an anonymous paper-and-pen questionnaire to collect data. The questionnaire comprised items on sociodemographic characteristics, use of the Internet and various platforms to seek health information. The Electronic Health Literacy Scale (eHEALS) questionnaire was applied to test electronic health (e-health) literacy.

Sociodemographic data comprised participants' age, type of school program (science–mathematics vs. humanities–languages), school performance as measured by the grade point average and the education level of parents. Education level of parents was classified as primary (≤8 years), secondary (9 – 12 years) and higher, that is, University (>12 years). Few parents had a primary level of education. Because of this, parents who had a primary and secondary education level were grouped as one category. School performance ranged from 2.0 to 5.0, because the school grading system regards Grade 2 as "pass" and Grade 5 as "excellent."

Further questions focused on the experiences with the Internet: whether the students used the Internet and at what age they were first exposed to the Internet. A study by Sebelefsky et al. (2015) provided the groundwork to define an array of web pages with various pieces of health-related information. The spectrum of web pages was discussed within the research team to adapt the list for the Serbian context. The final list of web pages included health forums, social media, health blogs, websites run by physicians, websites run by health institutions, health portals and YouTube. A health forum is defined as a web page containing a written conversation through online posts about health topics between members that is visible to all web page users. Social media refers to Facebook, Instagram or any other network or online community where users can exchange written or video materials at the time of study. This excluded TikTok, as it was not in use at the time of the study. A health blog is defined as a web page that contains posts about health content written by one or more individuals and is regularly updated with new posts. Websites run by physicians are defined as the Internet pages created by individual healthcare professionals to communicate a certain health-related topic to a wider lay audience. Websites run by health institutions are defined as the Internet pages created by professional healthcare organizations (such as hospitals, institutes, etc.) to inform the public about health-related topics. A health portal is defined as a web page offering articles on health issues that is accessed via a designated password. This meant that students had to be portal users who made their own credentials to access the content. YouTube is a website where videos are shared with the audience and have the possibility to share comments on videos. The definitions of these Internet web pages were communicated by the research team before the questionnaire distribution.

To better understand the interests of adolescents, a list of health-related topics was defined based on previous research (Kanuga and Rosenfeld, 2004; Martinović et al., 2023). Mental health was ranked as one of the top health topics of interest for adolescents. Other topics included psycho-active substances (PAS, defined as alcohol and/or illicit drugs), bullying, domestic violence, partner violence, medications, cigarettes/tobacco, nutrition, fitness and sex. Students were asked to circle those health topics that they were interested in and look for them on the Internet.

Because the Internet is an important resource of health-related information, having skills to navigate the content online is essential in efforts to optimize searches for adequate and reliable information. This is the reason why e-health literacy is critical in the modern digital landscape, especially for young Internet users. The e-health literacy refers to searching, understanding and evaluating health-related information on the Internet (Norman and Skinner, 2006a). E-health literacy was assessed using the eHEALS (Norman and Skinner, 2006b). The eHEALS has eight items that rate students' answers on a 5-point Likert scale from 1 to 5: 1 indicates "*strong disagreement*" with a given statement and 5 indicates "*strong agreement*" with a given statement. Eight rankings are added to a summary eHEALS score (minimum–maximum range: 8–40). A higher eHEALS score indicates a stronger e-health literacy level. The eHEALS items test the confidence in finding, using and evaluating online health information with the goal of reaching informed decisions.

The eHEALS has been translated to Serbian and back to English to align with the internationally accepted methodology for the translation and cultural adaptation of questionnaires. After reviewing translated versions, a final version was approved by the research team. The results of the psychometric testing in the adolescent population suggest that eHEALS has good reliability (Cronbach's $\alpha$ coefficient of 0.849) (Gazibara et al., 2019) and a single-factor structure.

### Outcome

Students were asked to answer the question about whether they used the Internet to seek information about mental health, such as depressive symptoms, anxiety, other mood-related problems, anger, attention deficit, eating disorders and the like. Possible answers were "yes" and "no." "Yes" was implied by circling the mental health topic on the list of health topics printed in the questionnaire. Otherwise, if the mental health topic was not circled, it was implied that the student did not browse that topic on the Internet, so the response "no" was assigned.

### Data analyses

The normality of distribution for the continuous variables was examined. None of the continuous variables showed a normal distribution. Therefore, Mann –Whitney test (nonparametric test) was applied to assess the difference between continuous variables (age, school performance, age at first Internet use and eHEALS score). The $\chi^2$-test was applied to assess the difference between categorical variables when the observation count per cell was >5. Fisher's exact test was applied in cases when the observation count per cell was <5 (browsing content about domestic violence).

The association of each variable with online information seeking about mental health was tested in an univariate model (i.e., unadjusted model, where the only independent variable was entered). Next, two multivariate logistic regression models were applied. In the first one, we tested the association of health blogs, health forums, social media, websites run by physicians, websites run by health institutions, health portals and YouTube with online information seeking about mental health. In the second one, we

examined the association of other health-related topics of interest for adolescents with online information seeking about mental health. Both models included sociodemographic covariates (gender, age, type of school program, school performance and parental education level), as well as variables related to digital literacy (age at first Internet use and eHEALS score). This classification of the independent variables into two models was performed to avoid overadjustment. In this way, it was possible to clearly observe the association of Internet websites and health topics that were also sought after besides mental health, with online information seeking about mental health.

Models were tested for multicollinearity. Multicollinearity is typically present when the variable inflation factor (VIF) > 2.0. None of the logistic regression models showed VIFs > 2.0. Therefore, multicollinearity was ruled out. The analyses were done in the Statistical Package for Social Sciences (SPSS Inc., Chicago, IL, USA), version 20. The level of probability of $p < 0.05$ was deemed statistically significant.

## Results

### Sample description

This study analyzed data from 702 high school students, of which 408 (58.1%) were girls and 294 (41.9%) were boys. The median age of the study sample was 17 years. The youngest students were 14 years old (26, 3.7%). The oldest students were 19 years old (6, 0.9%). Other age groups were relatively similarly represented in the study sample: 23.1% of 15-year-olds, 21.5% of 16-year-olds, 23.6% of 17-year-olds and 27.2% of 18-year-olds. Slightly more participants were enrolled in the science–mathematics program (391, 55.7%). Median school performance was 4.53 out of 5.0. Three-quarters of parents (526, 74.9%) had higher education attainment.

All 702 students included in the study were Internet users. The median age when students in the study sample began using the Internet was 10 years. The youngest age at first Internet exposure was 2 years, and the oldest at 17 years. The prevalence of seeking information about mental health in our study sample was 23.5% (165/702). Median eHEALS score was 26 out of 40.

Of all the Internet websites, most students used YouTube (185, 26.4%), health portals (166, 23.6%), health forums (148, 21%) and websites run by physicians (128, 18.2%). The least proportion of students used social media (59, 8.4%) and blogs (65, 9.3%).

Of all the listed health topics, most students browsed the Internet in search of information about fitness (368, 52.5%), nutrition (246, 35%), sex (224, 31.9%) and PAS (162, 23.1%). The least proportion of students browsed the Internet in search of information about domestic violence (8, 1.1%), partner violence (11, 1.6%) and bullying (28, 4.0%).

Sociodemographic characteristics, digital literacy, browsing online platforms and topics according to the history of online information seeking about mental health are presented in Table 1. Few characteristics differed between students who looked for information about mental health online and those who did not. Students who sought online health information about mental health had slightly lower school performance and eHEALS score compared to students who did not report seeking this information on the Internet. These students also more often browsed social media, health blogs, websites run by physicians and health institutions, as well as content about PAS, bullying, medications and cigarettes/tobacco. Students who did not seek online information about mental health were more often interested in online content about sex (Table 1).

### Internet websites as contributors to online information seeking about mental health

The results of the univariate and multivariate logistic regression model examining the association between the use of different Internet web pages and seeking information about mental health are presented in Table 2. In the univariate model, having lower school performance, lower e-health literacy and browsing health blogs and websites run by physicians and health institutions were associated with online information seeking about mental health. The multivariate model showed that having lower school performance, lower eHEALS score and browsing health blogs, social media and websites run by physicians and health institutions were independently associated with online information seeking about mental health among adolescents (Table 2).

### Health-related topics as contributors to online information seeking about mental health

The results of the univariate and multivariate logistic regression model examining the association between the interests in various health topics and seeking information about mental health are presented in Table 3. In the univariate model, having lower school performance, lower e-health literacy and interest in PAS, bullying, medications, cigarettes/tobacco and sex were associated with online information seeking about mental health. The multivariate model showed that being a girl, studying in the humanities–languages program, searching for online information about PAS, bullying and medications were independently associated with online information seeking about mental health among adolescents (Table 3).

## Discussion

This research showed that 23.5% of high school students reported searching for information about mental health on the Internet. Adolescents who had poorer school performance and poorer e-health literacy, and who read health blogs, social media and websites run by physicians and health institutions were more likely to seek online information about mental health. In addition, adolescent girls who studied a humanities–languages program and who also searched for data on PAS, bullying and medications were more likely to seek information about mental health on the Internet. Findings from this study provide cues as to the profile of high school students who look for information on mental health, their preferred websites where they find comprehensible data, as well as other health-related topics concomitant with searches on mental health. These pieces of information are valuable because they pinpoint the specific issues that impact mental health and well-being that are of interest to adolescents (Ridout and Campbell, 2018; Scott et al., 2022).

We observed that approximately one in four high school students sought online information about mental health. Data from youth services in Australia and the United Arab Emirates suggest that most contacts are related to emotional challenges (Watling et al., 2021) and mood disorders (Barbato et al., 2021). A study in the neighboring Croatia reported that of all mental health-related topics, adolescents most often need information about depression

**Table 1.** Sociodemographic characteristics, digital literacy, browsing online platforms and topics according to online health information seeking about mental problems

| Variable | Seeking online health information about mental problems | | P for difference |
| --- | --- | --- | --- |
| | Yes<br>N = 165<br>n (%) | No<br>N = 537<br>n (%) | |
| Gender | | | |
| Female | 98 (59.4) | 310 (57.7) | 0.704 |
| Male | 67 (40.6) | 227 (42.3) | |
| Age* | 17.0 (2.75) | 17.0 (3.0) | 0.723 |
| Age categories | | | |
| 14 years | 7 (4.2) | 19 (3.6) | 0.759 |
| 15 years | 34 (20.6) | 128 (23.8) | |
| 16 years | 38 (23.0) | 113 (21.0) | |
| 17 years | 38 (23.0) | 128 (23.8) | |
| 18 years | 47 (28.6) | 144 (26.9) | |
| 19 years | 1 (0.6) | 5 (0.9) | |
| Type of school program | | | |
| Science–mathematics | 85 (51.5) | 306 (57.0) | 0.210 |
| Humanities–languages | 80 (48.5) | 231 (43.0) | |
| School performance* | 4.50 (0.80) | 4.57 (1.0) | **0.017** |
| Parental education | | | |
| Primary and secondary | 41 (24.8) | 135 (25.1) | 0.940 |
| University | 124 (75.2) | 402 (74.9) | |
| Age at first internet use* | 10.0 (3.0) | 10.0 (4.0) | 0.343 |
| eHEALS score* | 25.0 (9.0) | 26.0 (9.5) | **0.016** |
| Browsing health forums | 37 (22.4) | 111 (20.7) | 0.629 |
| Browsing social media | 20 (12.1) | 39 (7.3) | **0.049** |
| Browsing health blogs | 26 (15.8) | 39 (7.3) | **0.001** |
| Browsing websites run by physicians | 40 (24.2) | 88 (16.4) | **0.022** |
| Browsing websites run by health institutions | 31 (18.8) | 63 (11.7) | **0.020** |
| Browsing health portals | 44 (26.7) | 122 (22.7) | 0.297 |
| Browsing YouTube | 52 (31.5) | 133 (24.8) | 0.085 |
| Browsing content about PAS | 82 (49.7) | 80 (14.9) | **0.001** |
| Browsing content about bullying | 14 (8.5) | 14 (2.6) | **0.001** |
| Browsing content about domestic violence | 3 (1.8) | 5 (0.9) | 0.400 |
| Browsing content about partner violence | 5 (3.0) | 6 (1.1) | 0.084 |
| Browsing content about medications | 46 (27.9) | 44 (8.2) | **0.001** |
| Browsing content about cigarettes/tobacco | 32 (19.4) | 34 (6.3) | **0.001** |
| Browsing content about fitness | 78 (47.3) | 290 (54.0) | 0.130 |
| Browsing content about nutrition | 50 (30.3) | 196 (36.5) | 0.145 |
| Browsing content about sex | 79 (7.9) | 145 (27.0) | **0.001** |

Note: Legend: eHEALS, Electronic Health Literacy Scale; PAS, psychoactive substances. *Median (interquartile range).

(26.2%), while interest in anorexia/bulimia, aggression, anxiety and self-harm was expressed by 10–12% of adolescents (Martinović et al., 2023). Because a considerable proportion of adolescents in this study reported having searched for information about mental health, addressing mental health in high schools through the initiatives of the Serbian government (Ministry of Health, 2019) and UNICEF (UNICEF Serbia, 2021) should be further enhanced to include active participation and dialogue to better understand what challenges adolescents encounter, with a special focus on destigmatization.

The study among Croatian adolescents found that information needs about depression, anorexia/bulimia, anxiety and self-harm were considerably higher among adolescent girls compared to boys (Martinović et al., 2023). In this study, we observed that girls were more likely to seek online information about mental health, which is in line with previous evidence about gender differences in overall health information seeking (Park and Kwon, 2018). Further evidence suggests that adolescent boys who more strongly conform to masculinity constructs are less inclined to seek mental health support (Clark et al., 2020). However, those adolescent boys who

**Table 2.** Internet platforms associated with online information seeking about mental health problems among high school students in Belgrade: Results of a multivariate model

| Variable | Univariate model | | Multivariate model | |
|---|---|---|---|---|
|  | OR (95% CI) | p | OR (95% CI) | p |
| Gender<br>  Girls vs. boys | 0.93 (0.65–1.33) | 0.704 | 0.94 (0.63–1.39) | 0.746 |
| Age | 1.02 (0.89–1.18) | 0.759 | 0.93 (0.79–1.10) | 0.388 |
| Type of school program<br>  Science vs. Humanities | 1.25 (0.88–1.77) | 0.217 | 1.29 (0.89–1.87) | 0.171 |
| School performance | 0.67 (0.50–0.91) | **0.011** | 0.61 (0.43–0.86) | **0.005** |
| Parental education<br>  Primary/secondary vs. University | 1.02 (0.68–1.52) | 0.940 | 1.04 (0.68–1.59) | 0.858 |
| Age at first internet use | 1.03 (0.96–1.11) | 0.376 | 1.05 (0.97–1.14) | 0.202 |
| eHEALS score | 0.97 (0.95–0.99) | **0.022** | 0.95 (0.93–0.98) | **0.001** |
| Browsing health forums | 1.11 (0.73–1.69) | 0.629 | 0.90 (0.56–1.43) | 0.650 |
| Browsing social media | 1.76 (0.99–3.11) | 0.052 | 1.95 (1.03–3.72) | **0.041** |
| Browsing health blogs | 2.39 (1.40–4.06) | **0.001** | 2.49 (1.39–4.46) | **0.002** |
| Browsing websites run by physicians | 1.63 (1.07–2.49) | **0.023** | 1.61 (1.01–2.58) | **0.047** |
| Browsing websites run by health institutions | 1.74 (1.09–2.79) | **0.021** | 2.01 (1.18–3.42) | **0.010** |
| Browsing health portals | 1.24 (0.83–1.84) | 0.297 | 1.25 (0.81–1.91) | 0.312 |
| Browsing YouTube | 1.40 (0.95–2.05) | 0.086 | 1.02 (0.65–1.58) | 0.943 |

*Note:* Bold values denote statistical significance.
Abbreviations: CI, confidence interval; eHEALS, Electronic Health Literacy Scale; OR, odds ratio; *p*, probability.

**Table 3.** Health-related topics associated with online information seeking about mental health problems among high school students in Belgrade: Results of a multivariate model

| Variable | Univariate model | | Multivariate model | |
|---|---|---|---|---|
|  | OR (95% CI) | p | OR (95% CI) | p |
| Gender<br>  Girls vs. boys | 0.93 (0.65–1.33) | 0.704 | 0.58 (0.36–0.92) | **0.022** |
| Age | 1.02 (0.89–1.18) | 0.759 | 0.85 (0.71–1.02) | 0.083 |
| Type of school program<br>  Science vs. Humanities | 1.25 (0.88–1.77) | 0.217 | 1.59 (1.05–2.39) | **0.027** |
| School performance | 0.67 (0.50–0.91) | **0.011** | 0.70 (0.48–1.03) | 0.073 |
| Parental education<br>  Primary/secondary vs. University | 1.02 (0.68–1.52) | 0.940 | 0.99 (0.62–1.59) | 0.991 |
| Age at first internet use | 1.03 (0.96–1.11) | 0.376 | 1.08 (0.99–1.18) | 0.069 |
| eHEALS score | 0.97 (0.95–0.99) | **0.022** | 0.98 (0.95–1.01) | 0.106 |
| Browsing content about psychoactive substances | 5.64 (3.83–8.31) | **0.001** | 4.09 (2.56–6.54) | **0.001** |
| Browsing content about bullying | 3.46 (1.62–7.42) | **0.001** | 2.48 (1.03–5.93) | **0.042** |
| Browsing content about domestic violence | 1.97 (0.47–8.33) | 0.357 | 2.36 (0.54–10.24) | 0.250 |
| Browsing content about partner violence | 2.77 (0.83–9.18) | 0.097 | 0.68 (0.12–3.85) | 0.664 |
| Browsing content about medications | 4.33 (2.74–6.86) | **0.001** | 4.68 (2.71–8.07) | **0.001** |
| Browsing content about cigarettes/tobacco | 3.56 (2.12–5.98) | **0.001** | 1.34 (0.70–2.56) | 0.377 |
| Browsing content about fitness | 0.76 (0.54–1.08) | 0.131 | 0.73 (0.46–1.15) | 0.170 |
| Browsing content about nutrition | 0.76 (0.52–1.10) | 0.145 | 0.80 (0.53–1.22) | 0.310 |
| Browsing content about sex | 2.48 (1.73–3.56) | **0.001** | 1.45 (0.89–2.36) | 0.135 |

*Note:* Bold values denote statistical significance.
Abbreviations: CI, confidence interval; eHEALS, Electronic Health Literacy Scale; OR, odds ratio; *p*, probability.

are less likely to seek help also tend to experience more psychological difficulties (Liddle et al., 2021). For this reason, improving mental health literacy of adolescent boys may be one of the major tasks in the efforts to promote adolescents' mental health.

Further characteristics relevant for online information seeking about mental health in this study were having lower school performance, studying humanities–languages program and having poorer self-reported e-health literacy. Lower academic achievement might be explained by the circumstance that adolescents who have sought information about mental health have certain mental health challenges that affected their school performance. However, we were unable to distinguish this feature more precisely as we have not included information about past mental health difficulties and visits to healthcare providers. Previous studies found that adolescents who need more health support encounter difficulties in school activities (Dalsgaard et al., 2020; López-López et al., 2021), which supports the notion that students who seek information about mental health might already be challenged by mental distress and, for that reason, are unable to score better at school.

High schools in Serbia provide general education and prepare students to enter university. However, students may be more inclined toward natural sciences or social sciences, so a division of the high school program has been made into a science–mathematics and humanities–languages program. Students must choose their preferred program at enrollment. The difference between the two programs is in the volume of classes: students in the science–mathematics program have more classes in chemistry, biology, physics and mathematics, while students in the humanities–languages program have more classes in history, philosophy, art and languages (two foreign languages plus Latin and more classes of literature). Usually, students in the university study a similar field to that of the high school program of their choice. The finding that students in the humanities–languages program were more likely to search for online information about mental health could be explained that this program has more classes in subjects that may touch upon mental health more often, such as sociology, philosophy and art, compared to natural sciences (Harackiewicz et al., 2016; Lavrijsen et al., 2021).

Higher levels of e-health literacy, on the other hand, have been associated with online information seeking in general (Park and Kwon, 2018). However, we observed that students who sought information about mental health online reported lower e-health literacy and may not be entirely certain which websites provide reliable information on the subject. As the sources of health-related information on the Internet are virtually endless, finding accurate information can be a difficult task for adolescents, especially because the regulation of online content about health is not yet satisfactory (Tonsaker et al., 2014). Bearing this in mind, enhancing overall e-health literacy in high school is in place through practical demonstrations and self-guided exercises.

Participants in this study used different Internet web pages to look for information about mental health. Some of them were formal and curated by health professionals, such as websites run by physicians and health institutions, and others were informal, such as health blogs or social media. It is not surprising that the use of formal sources was identified in this study as a contributor to online information seeking about mental health, and it is, in fact, encouraging that adolescents are browsing this content (Hanley et al., 2019; Murphy et al., 2020; Mancone et al., 2024). On these websites, adolescents can read edited and accurate data, as well as explanations about various illnesses and phenomena related to mental health. Once adolescents become familiar with mental

health vocabulary, symptoms and recommendations, they may have a greater sensitivity to identify potential initiation of mental health challenges within and around them and also seek help when needed (Özparlak et al., 2023).

Informal sources of online information about mental health may be more relatable to younger people, because they typically provide personal accounts of mental health challenges (Murphy et al., 2020; Wallström et al., 2021) that users can identify with and likely feel included or build empathy, or resilience. Previous reports aimed at mental health care providers suggest that blogs and social media were identified as useful channels to support mental health (Peek et al., 2015). An intervention to improve mental health that was based on blogging showed a significant improvement in well-being after 6 months of usage (Bickerstaff et al., 2021). Moreover, adolescents themselves articulated that social media could be a good platform for education about mental health (O'Reilly et al., 2019). These findings should be considered in efforts to set up online sources of information about mental health for adolescents.

A recent meta-analysis of 29 studies suggested that interventions in the digital realm have a compelling influence on the improvement of awareness about mental health disorders and improvement in mental health (Chen et al., 2024). While most adolescents, as digital natives, are well-versed with the Internet and digital devices, not all adolescents have a high level of digital health literacy, even in high-income countries (Taba et al., 2022; Stauch et al., 2025). In this context, low digital health literacy may be a major factor that undermines online promotion of mental health literacy, because of the risk of encountering misinformation, stigma-reinforcing content or inappropriate advice. Bearing these data in mind, having good digital health literacy is essential for novel modalities of learning and information dissemination. Thus, school-based digital health literacy programs, such as interactive workshops (Lewis et al., 2024), could play a key role in helping adolescents improve digital competencies related to health literacy that are critical for the optimization of online information seeking.

Interest in PAS, bullying and medications was associated with searching for online information about mental health, suggesting specific topics that adolescents relate to poor mental health. In fact, interest in PAS and medications was particularly strongly associated with seeking online information about mental health, because of the highest values of odds ratio, that is, adolescents who had interest in PAS and medications were four times more likely to browse the Internet in search of mental health information. The WHO highlights excessive alcohol intake and illicit drug use in adolescents as one of the key factors that affect mental health in adolescence (WHO, 2021). On the other hand, adolescents may be curious about mental health challenges associated with PAS use. Experiences and accounts of other young people who used PAS can be helpful in understanding of implications and long-term adverse effects of PAS use (Kara et al., 2023) and work as a type of "preventive intervention" for adolescents. Another issue regarding PAS is that adolescence is associated with risk-taking behaviors, and the first contact with PAS for many PAS users occurs in adolescence (Aly et al., 2020). Learning about the effects of PAS is closely tied to potential health-related consequences, including mental health decline. Thus, peer education and other modes of school interventions about PAS should also include the lived experiences of adolescents and young adults to better appreciate the impact on mental health.

Interest in online browsing about medications alongside mental health is another hallmark of information needs among adolescents,

as well as the tight connection between learning about medications and mental health. Medications are used to treat various mental health disorders; therefore, it is likely that adolescents hear about medications and their effects on mental health. Bearing in mind that depression and anxiety are the most common mental disorders in this population group (WHO, 2021), understanding the pharmacological treatment of these illnesses can provide baseline information about the advantages and side effects of prescribed medications. On the other hand, the use of medications as a means of psychoactive stimulation could be another explanation as to why the largest effect measures were identified for PAS and medications in relation to online information seeking about mental health. Based on these findings, interventions and programs focusing on raising awareness about mental health should also include information about the pros and cons of medication use to treat mental health disorders, as well as about the potential harmful effects of medications when they are being used inappropriately.

Exposure to bullying has also been known as the "risk for mental health" (Kaiser et al., 2020; WHO, 2021). In a nation-wide study of health behaviors among school children in Serbia, being bullied in secondary school was associated with more intense symptoms of depression (Skoric et al., 2023). Also, children who were bullies were more likely to drink alcohol (Santric-Milicevic et al., 2022). Bullying has been a long-standing issue in Serbian schools, and many initiatives have been undertaken to control it, and many adolescents are probably aware of this problem. As it is known that bullying strongly hurts adolescents' mental health, it is expected that high school students in this study had a keen interest in both topics.

It is important to highlight that online information seeking about mental health does not necessarily mean that those adolescents indeed have a mental health disorder or an immediate need for a professional intervention. Knowing about mental health disorders, their symptoms and associated behaviors is also important for early recognition and better understanding, as well as what to expect and how to behave around people who have mental health challenges, particularly when family and friends are affected (Migliorini et al., 2022). Mental health literacy has been in the spotlight over the past years, because the use of digital platforms to educate adolescents can have a strong impact on knowledge and attitude toward mental health disorders (Curran et al., 2023). It is especially relevant in efforts to reduce stigma about mental health and provide societal support for those who are challenged by mental disorders (Lanfredi et al., 2019). Ultimately, mental health literacy in adolescents free from mental health disorders can help optimize their mental health as well (Nobre et al., 2021).

This study has several limitations. All self-reported data, such as those in this study, are open to bias. This study was conducted at the beginning of 2017, and since that time, there have been changes in the online landscape, so our results may not entirely reflect the current situation in 2025. However, the majority of web pages included in this analysis are still being used today, particularly the web pages that are found as contributors to online mental health information seeking (websites run by physicians, websites run by institutions and health blogs) and can be a practical tool in efforts to increase mental health literacy in the population. We have not covered information about students' mental health, such as having previous or current mental health difficulties, nor did we examine health behaviors such as PAS use, involvement in bullying or use of specific medications. As a result, this analysis could not define whether adolescents who had previous mental health problems were more likely to search for online information about mental health. In terms of outcome, mental health problems covered numerous issues. Therefore, we were not able to distinguish whether adolescents had an interest in certain mental health problems over others. It should be kept in mind that participants in this study were recruited from the most urbanized setting in Serbia and are likely exposed to the Internet more frequently than their counterparts from rural settings. Therefore, the study results cannot be generalized to all adolescents in Serbia. Nevertheless, our future research goals are to compare urban and rural adolescents' digital consumer behaviors to better understand potential differences and information needs of young people in the contemporary digital landscape. Finally, the regression results were based on the cross-sectional methodology, which precludes inferences about causality.

One of the new digital platforms that were not in use at the time of this study was TikTok. Today, many adolescents prefer TikTok because its content is short and impactful, it can adapt to meet various demands and specific needs, because it is based on an algorithm that takes into consideration individual interests and preferences (McCashin et al., 2023). So, if an adolescent is browsing TikTok and other Internet pages about mental health, the algorithm will be offering further content related to previous searches. This means that adolescents who seek information about mental health would be additionally offered short videos about various aspects of mental health. In this way, adolescents could learn entirely new things, or be continuously provided with the content by the same creator (Basch et al., 2022). Information on TikTok is easily processed, so adolescents could follow and implement it in their everyday lives. It is indeed quite an influential digital platform that can have both positive and negative effects on its users (Chochol et al., 2023).

### Recommendations for practice and policy

This research provided evidence that almost one in every four adolescents has an interest in seeking information about mental health online. Adolescents associate mental health challenges with the use of psychoactive substances, bullying and medications, which should be prioritized in dialogues about mental health challenges. Formal (websites run by physicians and institutions) and informal online sources (social media and blogs) can support the integration of information and optimize comprehension of mental health matters. Being knowledgeable about mental health, free from stigma, is essential in fostering positive attitudes and approaches to mental health challenges. To help adolescents recognize the relevance of good mental health, parents, schools, youth organizations and governing structures need to work continuously to increase mental health literacy, address mental health challenges and provide suggestions on how to best improve mental health.

### Conclusion

In conclusion, this study suggests that 23.5% of high school students seek online information about mental health. These adolescents were more likely to be females, have lower school grades, study humanities–languages programs, have lower e-health literacy levels and use health blogs, social media and websites run by physicians and health institutions, as well as seek information about PAS, bullying and medications. Discourse about mental health in adolescents should include the aforementioned Internet platforms to raise awareness about mental health among adolescents and

improve their mental health literacy. It is of paramount importance to include adolescent boys in conversations about mental health. Adolescents have grown up using the Internet, so they will remain familiar with a variety of sources of online health information, but they still need guidance in terms of e-health literacy. Finally, discussions about mental health should also be part of high school education.

**Open peer review.** To view the open peer review materials for this article, please visit http://doi.org/10.1017/gmh.2025.10026.

**Data availability statement.** The dataset underlying this study is available on a reasonable request to the corresponding author.

**Acknowledgments.** The authors express their gratitude to all high school students who made this research possible.

**Author contribution.** TG contributed to the study design, analysis and interpretation of results and drafted the manuscript. JC and MC contributed to study design, data collection, analysis and interpretation of results and provided a critical review of the intellectual content of the manuscript. AG and TP contributed to the study design, analysis and interpretation of results and provided a critical review of the intellectual content of the manuscript. The authors agree to be held accountable for all aspects of the manuscript. The authors approved the final version of the manuscript before submission.

**Financial support.** This study was supported by the Ministry of Education, Science and Technological Development of the Republic of Serbia (Grant number 200100).

**Competing interests.** The authors declare none.

**Ethics statement.** Ethical approval to conduct the study was provided by the Institutional Review Board of the Faculty of Medicine, University of Belgrade (Approval number 747/I).

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
