## [Reviewer Report]

Overall, the manuscript is clear; however, the data presented is somewhat outdated, as the survey was conducted between December 2016 and 2017. The authors have acknowledged this as a study limitation. Below are a few suggestions for improvement:

Methods (Page 7, Paragraph 2): Enhance the explanation of school and student selection. Was the school selection process stratified or based on random sampling? How many schools were involved, and how many students were selected per school?

Results (Page 12, Paragraph 1): Include the response rate for better clarity.

Discussion (Page 14, Paragraph 1; Page 16, Paragraph 1; Page 17, Paragraph 1): Provide appropriate references to support the arguments and key points presented.

---

## [Reviewer Report]

The manuscript titled “Factors associated with online information seeking about mental health among high school students in Belgrade, Serbia” addresses an important and timely topic with high relevance for public health, adolescent development, and digital health literacy. The study is well designed, the methods are appropriate, and the discussion is largely balanced and insightful.

Strengths of the Manuscript:

• The background is comprehensive and contextualizes both the international and national (Serbian) landscape effectively.

• The rationale for the study is clearly stated, linking the growing interest in adolescent mental health with the role of digital media and online resources.

• Methodology is rigorous, with careful consideration of potential confounders and appropriate statistical analysis (e.g., use of non-parametric tests and multivariate logistic regression models, attention to multicollinearity).

• Results are clearly presented, with meaningful interpretation in the discussion.

• Practical recommendations are provided for educational institutions, parents, and policymakers.

• The limitations of the study are transparently acknowledged, particularly regarding the age of the data and the urban-only sample.

Areas for Improvement and Suggestions:

1. Distinction between information-seeking and help-seeking behavior:

While the introduction acknowledges that adolescents may seek information out of curiosity rather than personal distress, this distinction could be further emphasized throughout the discussion. It is important to stress that online information seeking does not necessarily imply the presence of a mental health disorder or an immediate need for professional intervention.

2. Expansion on digital health literacy implications:

Given that lower eHEALS scores were significantly associated with seeking mental health information online, the discussion could benefit from a deeper exploration of the implications of low digital health literacy. In particular, the risk of encountering misinformation, stigma-reinforcing content, or inappropriate advice could be highlighted. Suggestions for educational interventions (e.g., school-based digital health literacy programs) could strengthen the manuscript.

3. Interpretation of effect sizes:

The discussion could be enriched by more explicitly commenting on the strength of associations (e.g., odds ratios from the logistic regression models). For example, browsing for information on psychoactive substances and bullying were particularly strongly associated with mental health information seeking. This practical relevance should be highlighted.

4. Updating context regarding digital platforms:

Since the data collection occurred in 2016-2017, the discussion could better acknowledge the rise of newer platforms (e.g., TikTok) that are now highly influential among adolescents. Although the authors mention this limitation, providing a brief reflection on how the results might translate into today’s digital ecosystem would enhance the contemporary relevance of the findings.

5. Minor edits:

There are a few minor typographical errors throughout the manuscript (e.g., “instituttions” instead of “institutions”) which should be corrected during proofreading.

Final Recommendation:

Overall, this manuscript is a valuable contribution to the literature on adolescent mental health information behavior. I recommend minor revisions to enhance clarity, contextual relevance, and depth of discussion.

---

## [Reviewer Report]

I thank and commend the authors for taking the reviewers' comments seriously and thoughtfully incorporating them into their manuscript. In doing so, we are collectively contributing to the overall quality of both the individual articles and the journal as a whole.